# Assessment of dynamic cerebral autoregulation in humans: Is reproducibility dependent on blood pressure variability?

Jan Willem Elting[1]*, Marit L. Sanders[2], Ronney B. Panerai[3], Marcel Aries[4], Edson Bor-Seng-Shu[5], Alexander Caicedo[6], Max Chacon[7], Erik D. Gommer[8], Sabine Van Huffel[9], José L. Jara[7], Kyriaki Kostoglou[10], Adam Mahdi[11], Vasilis Z. Marmarelis[12], Georgios D. Mitsis[13], Martin Müller[14], Dragana Nikolic[15], Ricardo C. Nogueira[5], Stephen J. Payne[11], Corina Puppo[16], Dae C. Shin[12], David M. Simpson[15], Takashi Tarumi[17], Bernardo Yelicich[16], Rong Zhang[17], Jurgen A. H. R. Claassen[1]

1 Department of Neurology, University Medical Center Groningen, Groningen, The Netherlands, 2 Department of Geriatric Medicine, Radboudumc Alzheimer Centre and Donders Institute for Brain, Cognition and Behaviour, Radboud University Medical Center, Nijmegen, The Netherlands, 3 Department of Cardiovascular Sciences and Leicester Biomedical Research Centre in Cardiovascular Sciences, Glenfield Hospital, Leicester, United Kingdom, 4 Department of Intensive Care, University of Maastricht, Maastricht University Medical Center, Maastricht, The Netherlands, 5 Department of Neurology, Hospital das Clinicas University of Sao Paulo, Sao Paulo, Brazil, 6 Mathematics and Computer Science, Faculty of Natural Sciences and Mathematics, Universidad del Rosario, Bogotá, Colombia, 7 Departemento de Ingeniería Informática, Universidad de Santiago de Chile, Santiago de Chile, Chile, 8 Department of Clinical Neurophysiology, University of Maastricht, Maastricht University Medical Center, Maastricht, The Netherlands, 9 Department of Electronic Engineering, Stadius Center for Dynamical Systems, Signal Processing and Data Analytics, Katholieke Universiteit Leuven, Leuven, Belgium, 10 Department of Electrical, Computer and Software Engineering, McGill University, Montreal, Canada, 11 Department of Engineering Science, University of Oxford, Oxford, United Kingdom, 12 Department of Biomedical Engineering, University of Southern California, Los Angeles, California, United States of America, 13 Department of Bioengineering, McGill University, Montreal, Canada, 14 Department of Neurology, Luzerner Kantonsspital, Luzern, Switzerland, 15 Institute of Sound and Vibration Research, University of Southampton, Southampton, United Kingdom, 16 Departamento de Emergencia, Hospital de Clínicas, Universidad de la República, Montevideo, Uruguay, 17 The Institute for Exercise and Environmental Medicine, Presbyterian Hospital Dallas, University of Texas Southwestern Medical Center, Dallas, Texas, United States of America

* j.w.j.elting@umcg.nl

**Data Availability Statement:** All relevant data are available within the paper and its Supporting Information files.

## Abstract

We tested the influence of blood pressure variability on the reproducibility of dynamic cerebral autoregulation (DCA) estimates. Data were analyzed from the 2$^{nd}$ CARNet bootstrap initiative, where mean arterial blood pressure (MABP), cerebral blood flow velocity (CBFV) and end tidal $CO_2$ were measured twice in 75 healthy subjects. DCA was analyzed by 14 different centers with a variety of different analysis methods. Intraclass Correlation (ICC) values increased significantly when subjects with low power spectral density MABP (PSD-MABP) values were removed from the analysis for all gain, phase and autoregulation index (ARI) parameters. Gain in the low frequency band (LF) had the highest ICC, followed by phase LF and gain in the very low frequency band. No significant differences were found between analysis methods for gain parameters, but for phase and ARI parameters, significant differences between the analysis methods were found. Alternatively, the Spearman-Brown prediction formula indicated that prolongation of the measurement duration up to 35

**Funding:** The author(s) received no specific funding for this work.

**Competing interests:** The authors have declared that no competing interests exist.

minutes may be needed to achieve good reproducibility for some DCA parameters. We conclude that poor DCA reproducibility (ICC<0.4) can improve to good (ICC > 0.6) values when cases with low PSD-MABP are removed, and probably also when measurement duration is increased.

## Introduction

Dynamic cerebral autoregulation (DCA) is a key mechanism in cerebral homeostasis and protects the brain from alterations in blood pressure by arteriolar vasodilatation or constriction [1],[2]. Through this process, cerebral blood flow is preserved at a relatively constant level. Abnormal DCA status has been found in many pathological conditions such as stroke and traumatic brain injury[3],[4],[5],[6]. Despite detectable differences between healthy controls and patient groups, high variability is present which reduces the reliability of individual estimates[1]. Distributions of patient and healthy subject groups overlap considerably which results in poor diagnostic properties. This hampers implementation of DCA measurements as a diagnostic or monitoring tool in clinical practice.

The causes of variability in DCA measurements can be categorized as physiological or methodological. Physiological causes of DCA variability include the influence of confounders such as $PaCO_2$, autonomic nerve activity, and the amount of blood pressure variability[7],[8], [9]. Non stationarity of cerebral autoregulation suggests that DCA activity is not always constant over time, and can also be thought of as physiological variability[10],[11]. Non stationarity may be linked to variations in blood pressure variability or absolute blood pressure; if no change is present in blood pressure, no change in arteriolar diameter is needed to keep cerebral blood flow constant. Methodological causes of variability include measurement conditions and analytical techniques used to estimate DCA metrics. The influence of measurement conditions has been demonstrated previously: time of day, use of caffeine, amount of rest before measurement, body position, and external influences are all important to consider[12]. Recommendations for optimal settings of these components have been published in the 1st CARNet bootstrap initiative[12]. Many different analysis strategies for DCA assessment have been developed, some of which try to model and account for physiological causes of variability, such as $PaCO_2$. For example, Transfer Function Analysis (TFA) based on multiple inputs (MABP, $CO_2$) allows for correction of $CO_2$ influences[13],[14], while Volterra Kernels[7],[15] or Multi-model Pressure Flow Modeling[16],[17] try to capture the non-linear aspects of cerebral autoregulation. Time varying autoregressive moving average models (ARMA models) and multiple-input finite impulse response models have been used to model the non-stationary aspects of cerebral autoregulation[18],[19],[20]. However, at present it is not clear if any of these methods is superior to others in terms of variability and reproducibility of the DCA estimates.

The second CARNet bootstrap initiative was started in 2015 to gain further insight into the problem of variability of DCA estimates. Specifically, reproducibility was assessed to quantify variability between repeated measurements. The main focus of this study was to evaluate any differences in reproducibility between the different analysis strategies that have been developed. Initially, surrogate data were used to isolate the separate effects of modelling method from physiological variability[21]. In the second analysis, physiological data were used to quantify reproducibility of several DCA analysis methods on physiological data[22]. The third analysis is described in this study, focusing on the effect of blood pressure variability,

quantified as power spectral density of mean arterial blood pressure (PSD-MABP), on DCA reproducibility. Some studies have shown that DCA parameters are more stable if PSD-MABP is high, and can become variable if PSD-MABP is low[23],[24]. A number of techniques to increase PSD-MABP variability have been described, and in some of these studies reproducibility of the DCA estimates have been shown to improve[23],[25],[26],[27].

The specific hypothesis that were tested are:

1. DCA Reproducibility will increase if subjects with low PSD-MABP are left out of the analysis.

2. DCA Reproducibility after PSD-MABP based case removal is similar for different analysis methods.

3. DCA Reproducibility after PSD-MABP based case removal is similar for different DCA parameters.

## Materials and methods

### Subjects and centers

S1 and S2 Tables summarize the participating centers, their roles and the analysis methods. Measurements from 75 subjects were collected from 6 participating centers. A formal sample size calculation was not possible before the study was performed, due to the lack of information about inter-subject variability for the majority of autoregulation indices. In a previous study, Brodie et al. provided estimates for the sample size in studies using the ARI index, but similar information was not available for the other indices in our study. From the information provided, with n = 75 subjects in our case, we can estimate that a difference of ARI = 0.77 could be detected with 80% power at $\alpha = 0.05$, which should be very satisfactory to detect intra-subject differences [28].

Subjects were 18 years or older, male or female, and were in good health. Exclusion criteria included a history of uncontrolled hypertension, smoking, diabetes, irregular heart rhythm, cardiovascular disease, TIA/stroke or significant pulmonary disease. The study has been carried out in accordance with the Code of Ethics of the World Medical Association (Declaration of Helsinki). Written informed consent was obtained from all subjects. A study flowchart is presented in Fig 1.

### Measurements

Measurement data had to consist of transcranial Doppler ultrasound (TCD) derived cerebral blood flow velocity in the middle cerebral artery (CBFV) data and MABP as well as end tidal $CO_2$ (EtCO2) data. TCD recordings could be unilateral or bilateral. Beat to beat data had to be resampled at 10 Hz. For each subject, 2 periods of at least 5 minutes of continuous, good quality, artifact free data were required. The interval between the measurements could not exceed 3 months. Subjects were measured supine at rest. No MABP oscillation inducing maneuvers were performed.

### Pre-analysis data validation

Data validation was performed using the following criteria:

Mean CBFV > 30 cm/s, $EtCO_2$ > 3 kPA and < 7 kPA, and mean MABP >40 and < 160 mmHg at all times. Further visual inspection of all data consisted of checking for step artifacts, and other artifacts such as large slow drifts. Based on these criteria, 6 subjects were excluded

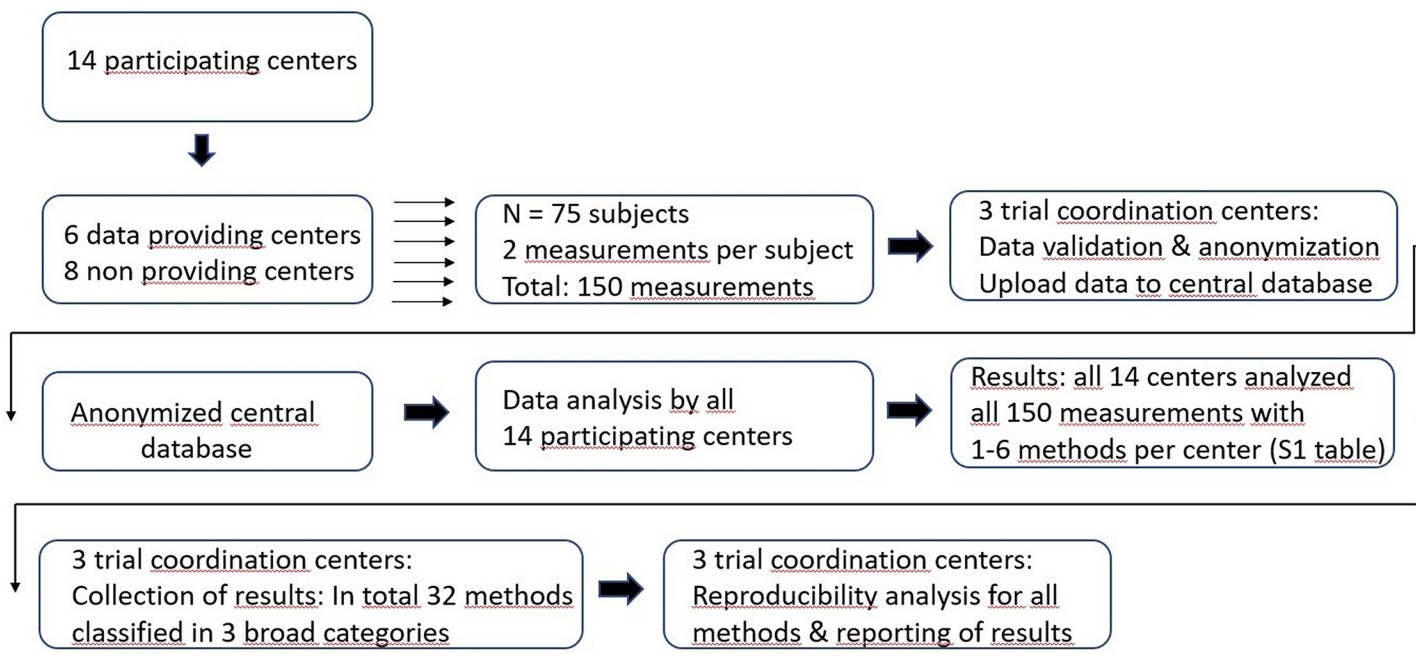

**Fig 1. Study flowchart.**

from the analysis, and were replaced with good quality backup data from 1 center. Missing data from one hemisphere (CBFV), were replaced with artificially generated data. This procedure resulted in N = 55 left sides, N = 71 right sides, and N = 22 artificial sides. The results for the artificial and physiological data have been presented elsewhere, and will not be reported here[21]. Since there were no significant differences between left and right sided values for all DCA variables examined in this study, left and right sides were averaged in case of bilateral measurement, or were used singly in case of unilateral missing data.

## Analysis methods

S1 Table summarizes the analysis methods that were used by each center. The full details of all analysis methods will not be presented here, instead references for each method are provided. Each method was categorized into one of three possible broad categories of methods: 1. Transfer function like methods: those that produce at least some form of phase and/or gain estimates. 2. Autoregulation index like methods: Those that produce an autoregulation index which ranges from 0–9. 3. Correlation index like methods: Those that produce a correlation like index. These categories were created from the perspective of similar output parameters, not because the analysis is similar on mathematical grounds.

## PSD-MABP based case removal

Power spectra were created by calculating power spectral density (PSD) of the mean arterial blood pressure (MABP) signals. This was not part of the analysis of the individual centers, instead it was performed centrally. A Hanning window was applied, and segments of 100 seconds with 50% overlap were used, according to the Welch method of spectral estimation. Mean PSD-MABP estimates were calculated in VLF (0.02–0.07 Hz) and LF (0.07–0.2 Hz). Histograms of the PSD-MABP estimates were created and 10 cut-off levels were defined, which reflected the 0, 10, 20, 30, 40, 50, 60, 70, 80 and 90th percentiles of the PSD-MABP distribution.

Each cut-off level was used to remove cases with PSD-MABP levels below these values. The same cut-off levels were used for all analysis methods, which results in the same number of cases for each analysis method for each cut-off level. For DCA variables, cases were removed based on the corresponding frequency band PSD-MABP (DCA-VLF vs PSD-MABP-VLF and DCA-LF vs PSD-MABP-LF) but for DCA variables in the LF band cases were also removed using PSD-MABP in the VLF.

## Statistical analysis

To evaluate the possible impact of PSD-MABP levels on DCA variables, the relation between the absolute difference of the duplicate DCA variable measurements and the lowest PSD value of the PSD-MABP signal of both measurements was analyzed using Spearman rank correlation for each method. We used the absolute difference for this analysis, since it is irrelevant which of the two measurement has the higher or lower value of the DCA variable. The lowest PSD value was used, since this will assure that when using PSD-MABP as a filter variable based on threshold levels, both PSD-MABP values will be above the threshold. To investigate main effects, the Spearman correlation coefficients were used as a summary measure and were tested against the null hypothesis value of zero using a one sample T test.

To assess reproducibility, one way Intraclass Correlation Coefficient (ICC) analysis was performed on all available data, and for each subgroup created through case removal based on PSD-MABP. The one way setting in SPSS, version 22, was used to calculate the ICC. To meet the prerequisite of bivariate normal distributions, data were transformed using Box-Cox transformations[29]. Within one analysis method, the same transformation was applied to both the first and second measurement, but different transformations could be used for different methods and DCA variables. To quantify the effect of case removal based on PSD-MABP on ICC values for different DCA variables within the main method groups, a repeated measurements ANOVA was used, entering the ICC values at the 10 PSD levels as repeated measurements and DCA variable type as the group variable. Gain VLF, Gain LF, Phase VLF, Phase LF and ARI were the possible values for the group variable. Post Hoc tests were performed to assess differences between the DCA variable groups.

To test for differences in ICC values between different analysis methods, Monte Carlo simulations were conducted using correlated random data with an inhouse written script in LabVIEW 2014[30]. The details of the Monte Carlo simulations can be found in S1 File.

A final analysis consisted of extrapolating the ICC result to a longer measurement period. The Spearman-Brown prophecy formula was used to predict the length of time needed for a DCA variable to reach a good ICC value of at least 0.6[31]. The estimated longer durations involve the assumption that autoregulation would remain stable when the measurement duration is extended. The Spearman-Brown calculated ICC ($ICC^{SB}$) is calculated as:

$ICC^{SB} = \frac{nICC}{1+(n-1)ICC}$ where ICC is the ICC value based on 2 measurements of 5 minutes and $n$ is the factor by which the measurement duration will be extended.

## Results

### Subject characteristics

Thirty-three subjects (44%) were female and 42 were male. Mean age was 47.8 ± 18.6 years (range 20–80). Five subjects had a history of asymptomatic hypertension, which was well controlled by medication at the time of measurement. Median lowest PSD-MABP was comparable between those that used vasoactive medication and those without any medication (S1 Fig)

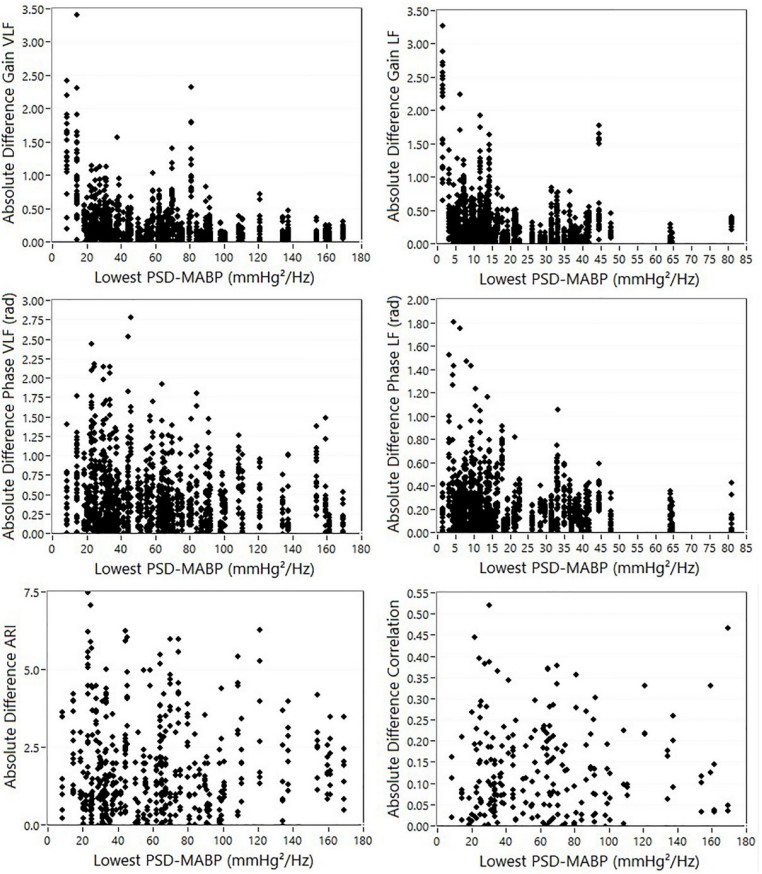

**Fig 2. Absolute between measurement differences vs the lowest PSD-MABP of repeated measurements.**

Four older subjects had a history of mild cognitive impairment, but not dementia. Another 4 subjects used NSAIDs regularly, without any history of vascular disease.

## Correlation between DCA variables and PSD-MABP

Fig 2 shows the correlation between the absolute difference of DCA variables between the first and second measurement and the lowest PSD-MABP value of both measurements. High absolute differences tend to occur at low PSD-MABP values, especially for gain variables, and to a lesser degree also for phase variables. For ARI and correlation like variables, this dependency is less clear or absent. Results of the Spearman rank correlation analysis can be found in Table 1. For individual methods, differences in gain in the VLF had a strong relation to the PSD-MABP estimates; for other variables, this relationship was less strong. When all Spearman correlation coefficients were used as summary measures, mean gain (VLF: -0.25 ± 0.1, p<0.001; LF: -0.19 ± 0.06, p<0.001) and phase (VLF: -0.13 ± 0.09, p<0.001; LF: -0.11 ± 0.08, p<0.001) coefficients were significantly different from zero. However, for ARI (0.00 ± 0.11, p = 0.98) and correlation like indices (0.05 ± 0.02, p = 0.10), no significant correlation was found.

## Reproducibility: ICC analysis for incremental case removal

Figs 3–5, and S2–S4 Figs summarize the main findings of the ICC analysis for different levels of PSD-MABP based case removal. For gain, phase and ARI, mild to moderate increases in

**Table 1. Spearman correlation coefficients for the absolute DCA variable difference between the two measurements and the lowest PSD-MABP value of both measurements.**

| Method | Category | Gain VLF | Gain LF | Phase VLF | Phase LF | ARI | Cor |
|---|---|---|---|---|---|---|---|
| 1.1:TFA | 1 | -0.33* | -0.08 | -0.13 | -0.18 | | |
| 1.2:ARI | 2 | | | | | -0.00 | |
| 2.1:V.Kernels.SISO | 1 | -0.17 | -0.23* | -0.19 | -0.13 | | |
| 2.2:V.Kernels.MISO | 1 | -0.08 | -0.23* | -0.28* | -0.03 | | |
| 3.1:TFA | 1 | -0.28* | -0.12 | -0.03 | -0.09 | | |
| 3.2:TFA | 1 | -0.25 | -0.15 | | | | |
| 4.1:ARI:FFT | 2 | | | | | -0.02 | |
| 4.2:ARI:MA1 | 2 | | | | | 0.13 | |
| 4.3:ARI:MA2 | 2 | | | | | 0.15 | |
| 5.1:TFA | 1 | -0.39* | -0.19 | -0.25* | -0.19 | | |
| 5.2: SIDE-ObSP | 3 | | | | | | 0.06 |
| 6.1:TFA | 1 | -0.1 | -0.15 | -0.06 | -0.27* | | |
| 7.1:TFA | 1 | -0.21 | -0.18 | -0.24* | -0.14 | | |
| 8.1:ARX | 2 | | | | | -0.11 | |
| 8.2:Wavelets | 1 | | | -0.05 | -0.16 | | |
| 9.1:TFA | 1 | -0.29* | -0.25* | -0.15 | -0.16 | | |
| 9.2:CCM | 3 | | | | | | 0.03 |
| 11.1:TFA | 1 | -0.28* | -0.13 | -0.15 | -0.04 | | |
| 11.2:TFA | 1 | -0.25* | -0.17 | -0.15 | -0.04 | | |
| 11.3:TFA | 1 | -0.21 | -0.15 | | | | |
| 11.4:TFA | 1 | | -0.17 | | 0.03 | | |
| 11.5:IR coeff | 2 | | | | | -0.19 | |
| 11.6:TFA:MISO | 1 | | -0.15 | | -0.06 | | |
| 12.1:TFA | 1 | -0.42* | -0.22 | -0.09 | -0.15 | | |
| 12.2:ARI | 2 | | | | | -0.04 | |
| 12.3:Wavelets | 1 | -0.25* | -0.22 | 0.04 | -0.05 | | |
| 13.1:TFA | 1 | -0.27* | -0.06 | -0.10 | -0.17 | | |
| 14.1:ARX:SISO | 1 | -0.25* | -0.22 | -0.09 | 0.02 | | |
| 14.2:ARX:MISO | 1 | -0.36* | -0.25* | -0.24* | -0.03 | | |
| 14.3:FIR:SISO | 1 | -0.07 | -0.30* | -0.12 | -0.10 | | |
| 14.4:FIR:MISO | 1 | -0.2 | -0.32* | -0.03 | -0.06 | | |
| 14.5:TFA | 1 | -0.34* | -0.15 | -0.19 | -0.26* | | |
| Mean ± SD | | -0.25 ± 0.1 | -0.19 ± 0.06 | -0.13 ± 0.09 | -0.11 ± 0.08 | 0.00 ± 0.11 | 0.05 ± 0.02 |
| Probability = 0 | | < 0.001 | < 0.001 | < 0.001 | < 0.001 | 0.98 | NA |

* = Significant correlation, p<0.05.

VLF = Very low Frequency, LF = Low Frequency. Category: 1 = TFA, 2 = ARI, 3 = correlation. SISO = Single Input (MABP), Single Output (CBFV). MISO = Multiple Input (MABP, $EtCO_2$), Single Output (CBFV). Note that especially gain in the VLF is strongly correlated with PSD-MABP. For ARI and correlation like indices, no significant correlation was found. The last two rows show the mean of all Spearman coefficients, and the result of the one sample T test vs value 0.

ICC can be seen, while for correlation like indices, no clear change in ICC is present. There was a significant linear increase in ICC with increasing numbers of cases removed based on PSD-MABP for gain (VLF: F = 34.3 p<0.001, LF: F = 8.6 p = 0.008) and phase (VLF: F = 15.2 p = 0.001, LF: F = 112.4 p<0.001), but not for ARI (F = 0.2 p = 0.68). For ARI, the absence was mainly due to a sudden decrease in ICC at the highest PSD-MABP cut-off level; when repeating the analysis without the last level, a significant linear increase in ICC was found (F = 9.7 p = 0.014). For gain LF, the increase in ICC was highest when removing cases with

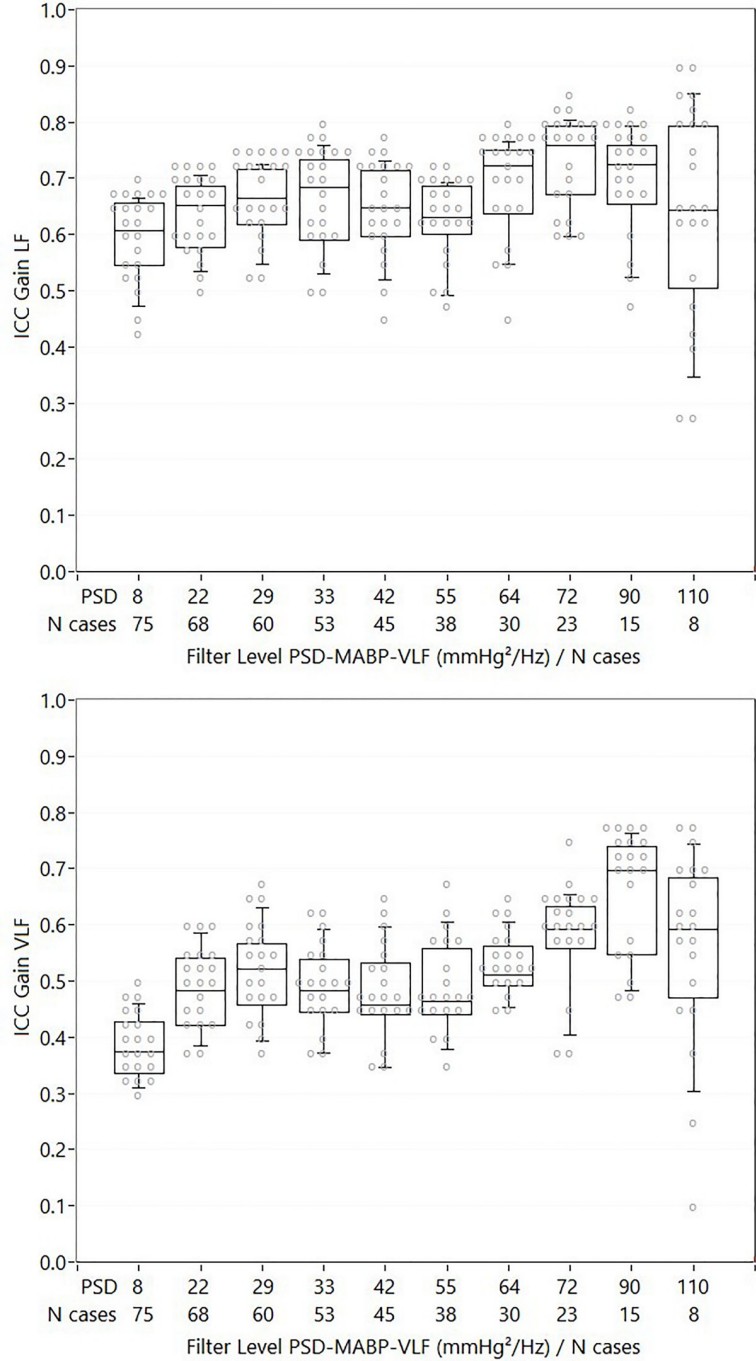

**Fig 3. ICC values for gain LF and gain VLF.** Beeswarm boxplot with ICC values for gain LF (upper figure) and gain VLF (lower figure) for different cut-off levels of PSD-MABP. Each grey dot represents an analysis method. No significant differences between methods were found. The increase in ICC with increasing cut-off levels was significant for both gain LF and gain VLF.

PSD-MABP-VLF, and not with PSD-MABP-LF. For all other DCA variables, the ICC increase was always higher when removing cases using the corresponding PSD-MABP frequency band. Viewed from the level of group averages, gain LF had significantly higher ICCs compared to

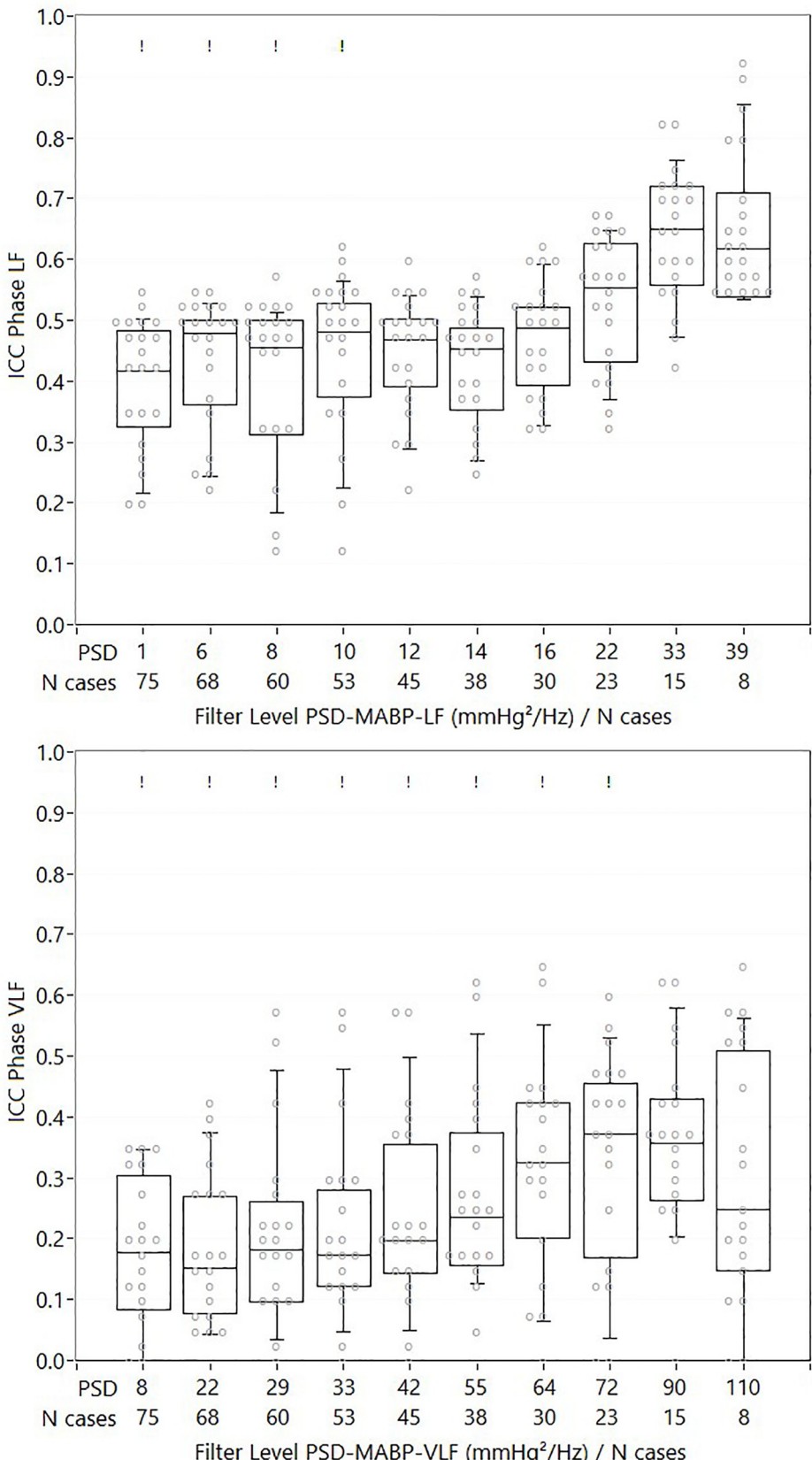

**Fig 4. ICC values for phase LF and phase VLF.** Beeswarm boxplot with ICC values for phase LF (upper figure) and phase VLF (lower figure) for different cut-off levels of PSD-MABP. Each grey dot represents an analysis method. ! :

indicates cut-off level at which significant differences in ICC values between the methods were found. The increase in ICC with increasing cut-off levels was significant for both phase LF and phase VLF.

all other DCA variable groups (p<0.001 for all comparisons), while gain VLF and phase LF had higher ICCs compared to phase VLF (both p<0.001). A significant interaction between PSD-MABP based case removal and DCA variable group was found (F = 4.1 p = 0.004).

Within each main method group, Monte Carlo simulations indicated significant differences between individual methods for phase VLF, phase LF and ARI, but not for gain VLF and LF. S2–S4 Figs show beeswarm letter-box graphs with post-hoc sum-scores for each individual method. For phase VLF, methods 14.1 and 14.2 (S1 Table, both ARX (autoregression) models with 1 and 2 inputs (MABP, MABP and CO2) respectively) were the methods that showed the clearest increase in ICC, with ICC values compatible with moderate to good reproducibility. For phase LF, multiple methods had ICC values in the moderate range, while some had very low ICC values. For ARI, method 11.5 (impulse response filter coefficient) had the highest reproducibility, with ICC values showing good reproducibility at low PSD-MABP cut-off levels.

## Spearman-Brown prediction of ICC values

Fig 6 shows the Spearman-Brown predicted ICC values for the median DCA variable values without PSD-MABP based case removal. To achieve a good ICC value of at least 0.6, the recording length must be increased to 10 minutes for median phase LF, 15 minutes for gain VLF, 20 minutes for ARI and 35 minutes for phase VLF. For gain LF, 5 minutes is sufficient.

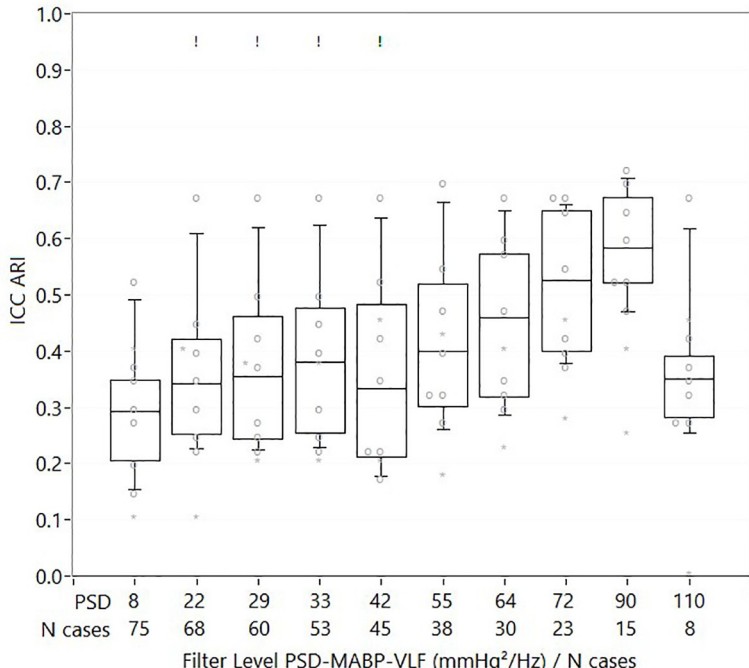

**Fig 5. ICC values for ARI and correlation like indices.** Beeswarm boxplot with ICC values for ARI and correlation like indices for different cut-off levels of PSD-MABP. Each grey dot represents an analysis method. ! : indicates cut-off level at which significant differences between the methods were found. * indicates correlation like indices, which were not included in calculation of the boxplot. The increase in ICC with increasing cut-off levels was significant for ARI like indices, when the highest cut-off level (110) was excluded.

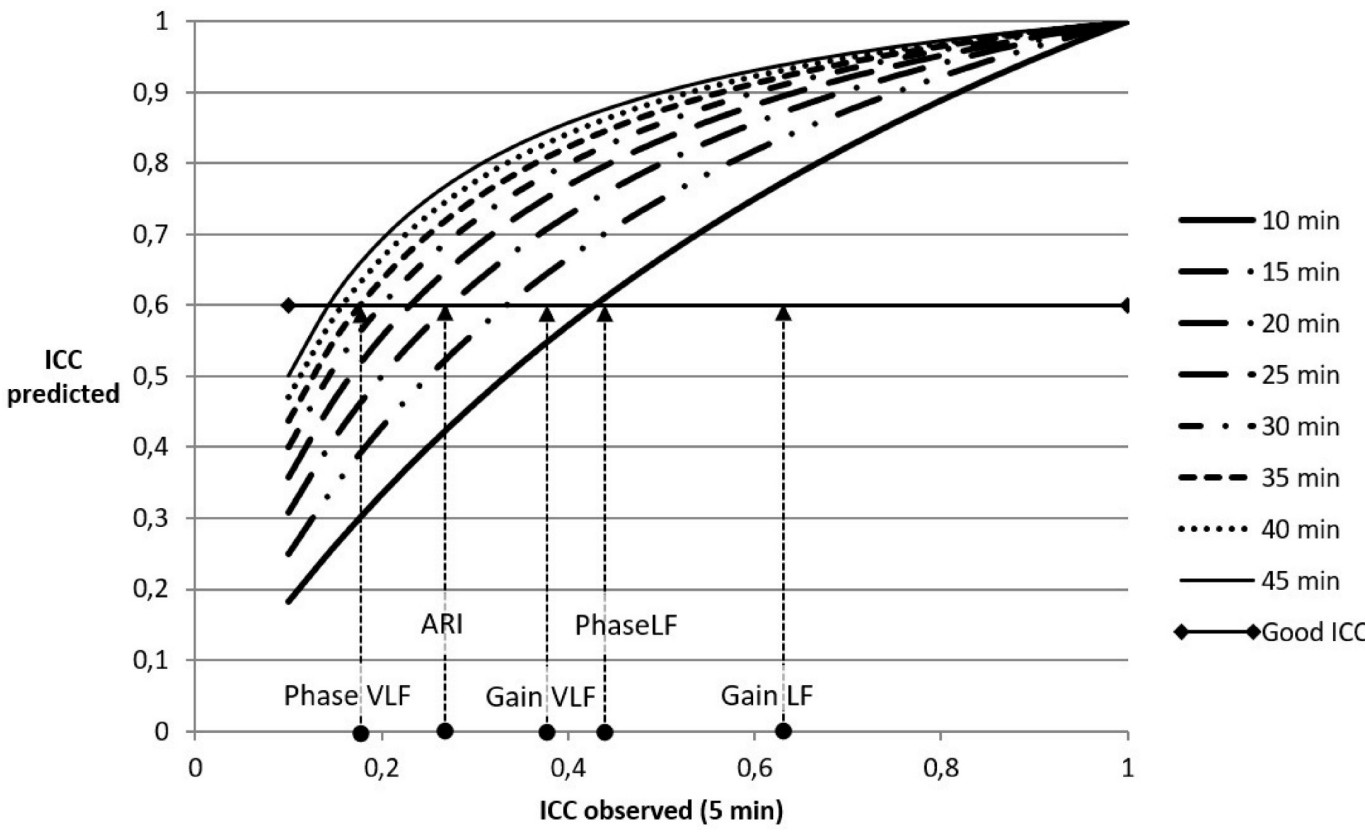

**Fig 6. Results of the Spearman-Brown analysis.** Spearman-Brown predicted ICC values based on single measures ICC values calculated on 2 measurement periods of 5 minutes. The assumption is that autoregulation would remain stable when the measurement duration is extended. The observed median ICC values for DCA variables based on all cases (n = 75) are projected onto the predicted ICC = 0.6 line. The first intersecting curve on the left represents the recording time that is needed to achieve an ICC of at least 0.6. To achieve an ICC value of ≥ 0.6 the recording length must be increased to 10 minutes for median phase LF, 15 minutes for gain VLF, 20 minutes for ARI and 35 minutes for phase VLF. For gain LF, 5 minutes is sufficient.

## Discussion

In this study, the effect of PSD-MABP on DCA reproducibility was investigated. In the first step, a dependency of reproducibility on PSD-MABP was established by correlating the absolute within subject difference of DCA variable values with the lowest PSD-MABP value of the two measurements. Significant correlations were exclusively found for TFA like DCA variables, with the highest correlations for gain. For ARI and correlation like methods, no significant correlations were found. This means that the effect of any case removal based on PSD-MABP levels on reproducibility can be expected to be higher in TFA like methods. However, absence of significant correlations does not mean that there can be no benefit in reproducibility following case removals: if only a few outliers exist at low PSD-MABP power, there may not be a significant correlation, but reproducibility statistics such as ICC values can still be significantly affected, as they are quite sensitive to even a few outliers. In line with this reasoning, increases in ICC after incremental case removal were most pronounced in DCA variables whose difference values were significantly correlated to PSD-MABP (gain and phase), but were also found in ARI like methods. For most DCA variables, case removal based on PSD-MABP within the corresponding frequency band yielded the highest increases in ICC, except for gain LF where case removal based on PSD-MAPB-VLF (instead of PSD-MABP-LF) resulted in the highest increases in ICC. The reason for this is not clear from these results, but

we speculate that the higher MABP oscillations in the VLF occur simultaneously with lower oscillations in the LF band, which may create a cross-dependency between LF band DCA parameters and PSD-MABP-VLF.

When interpreting the absolute and maximal values of ICC, and comparing the values with often quoted guidelines for the interpretation of ICC values (less than 0.40: poor, between 0.40 and 0.59: fair, between 0.60 and 0.74: good, between 0.75 and 1.00: excellent)[32], it is clear that for most DCA variables ICC is in the poor to fair range without any PSD-MABP based case removal, with the exception of gain LF. On average, gain LF was clearly the most reproducible DCA parameter, with significantly higher ICCs compared to all other DCA variables, followed by gain VLF and phase LF. With case removal, maximal ICC's can be in the range regarded as 'good' ($>$0.6), especially for TFA like variables such as gain, and to a lesser degree also phase. However, the maximum level of reproducibility that can be reached depends on the method that was used, as there were significant differences between the methods, mostly for phase and ARI. Noteworthy in this regard, is that for phase in the VLF, there does seem to be some benefit in using linear auto-regressive models, as methods 14.1 and 14.2 (S1 Table) have the highest ICCs in the VLF. Furthermore, for ARI like indices, the use of impulse response FIR filters seems advisable, given the high reproducibility for method 11.5 (impulse response filter coefficient).

What can be advised for the analysis of DCA in practical terms given these results? The answer to this question is not straightforward due to the variety of methods that were employed in this study, but will also depend highly on the type of clinical problem that is being addressed. As has been stated previously, it is unlikely that a single analysis method will prove to be satisfactory for all possible clinical situations[33,34]. Reproducibility by itself is not sufficient for good diagnostic properties of DCA variables, but it can be viewed as a necessary precondition for any diagnostic test. Therefore, the variable with the highest reproducibility (Gain LF), may not be necessarily the best variable to detect disease states, so we cannot recommend a preferred analysis method or DCA variable based on this study alone. However, it is unlikely that those methods with low reproducibility will be clinically useful. Increases in ICC after PSD-MABP based case removal are clearly present, but they are only achieved at the cost of leaving out a substantial amount of cases. These positive and negative aspects need to be carefully balanced in each clinical situation. One way to avoid the negative effects is to put further effort in investigating the influence of maneuvers that induce MABP oscillations[33]. The other option is to increase the duration of measurement in order to obtain more stable DCA estimates. Both from the viewpoint of DCA variables themselves and from the viewpoint of reproducibility statistics this may lead to more reliable results. For example, with TFA, it is known that the confidence limits of the estimates depend on the level of coherence, and on the number of degrees of freedom used in the estimate which is determined by measurement duration and spectral smoothing[35],[36]. With increasing measurement duration, confidence limits of DCA estimates will be reduced, leading to less variability and higher reproducibility. From the viewpoint of ICC, the Spearman-Brown formula was used to visualize that 5 minutes of measurement duration is not enough for most DCA metrics, except for gain LF. It is estimated that for some methods, extending the measurement duration to beyond 30 minutes will be required to achieve acceptable reproducibility. It needs to be emphasized that this is only a prediction based on the assumption of stable and stationary autoregulation, which may not always be realistic. Ideally, the prediction results should be verified in measurements of longer duration, for example the study of Zhang and colleagues have used measurement durations of up to 2 hours[37].

Clinical perspectives and recommendations:

Based on the results in this study, it seems advisable to calculate PSD-MABP in the VLF and LF bands during data acquisition, so that one can evaluate if sufficient power is present in the MABP signal. It is advisable to have at least 29 mmHg$^2$/Hz PSD in the VLF when analyzing gain and phase and also ARI, so that at least moderate reproducibility can be achieved for some methods, with good reproducibility for the best performing methods. This will be accompanied by a 20% loss of analyzable cases, if one extrapolates from the relatively large sample adopted in this study. When phase LF is of interest, at least 10 mmHg$^2$/Hz PSD is advisable, which will lead to 30% loss of cases, but these estimates need further confirmation in different measurement conditions and patient sub-groups. If not sufficient, recordings may be extended while selectively leaving out data segments with low PSD-MABP. Alternatively, oscillation inducing maneuvers can be attempted. In clinical situations, longer measurement duration without loss of data quality may sometimes be difficult, for example in acute stroke patients who may suffer from delirium and restlessness. On the other hand, in the intensive care setting when patients are sedated and intubated, a longer measurement duration may be more easily obtained.

A number of limitations need to be mentioned. First, this was a retrospective study, with highly selected measurements that were further inspected and checked for quality issues, and if necessary were replaced. The results that were found here may not always apply in situations where optimal recordings are not easily achieved. Secondly, some of the subjects were using medication during the time of the measurement, most often for hypertension. We cannot exclude the possibility that low PSD-MABP power was in part due to the use of medication, but suspect that it is likely that this is not a major cause, since median lowest PSD-MABP was comparable between those that used vasoactive medication and those without any medication (S1 Fig). Third, subjects of all ages were included, and this may have contributed to increased variability in some of the parameters. For DCA variables on the other hand, no clear influence of age has been established so far[38], therefore a major influence of age variability on the results seems unlikely. Fourth, only two centers employed correlation based methods in this study, therefore these methods could not be analyzed with the same level of detail as other methods. Lastly, we did not consider the influence of time between the two measurements, which could be up to 3 months in this study. We cannot exclude the possibility that a longer interval between the two measurements will introduce extra variability in the DCA estimates.

To conclude, the reproducibility of DCA parameters, derived from spontaneous fluctuations in MABP, is poor to moderate when all cases were used for the analysis, except for gain LF which has good reproducibility with TFA-like methods in some centers. By removing cases with low PSD-MABP power, reproducibility can increase to levels generally accepted as 'good', but only for a limited number of analysis methods. Future research should perform a similar assessment of reproducibility using protocols that induce PSD-MABP oscillations or use longer measurement durations at rest. Studies of short term reproducibility in patient-groups or induced pathological states would also be highly relevant to allow further progress with translation of assessment of dynamic CA methods to clinical practice.

## Supporting information

**S1 Table. Analysis methods with references and corresponding output variables per center.** Category: 1 = TFA-like methods, 2 = ARI-like methods, 3 = correlation-like methods, Method group: 1 = TFA, 2 = Laguerre expansions, 3 = Wavelets, 4 = IR-filter, 5 = ARX, 6 = ARI, 7 = ARMA-ARI/ARX, 9 = IR-filter, 10 = correlation coefficient; VLF: very low frequency; LF: low frequency; BP: blood pressure; FFT: fast Fourier transform; ARI: autoregulation index; ARX:

autoregressive model with exogenous input; Centre names are listed in S2 Table.
(DOCX)

**S2 Table. Participating centers and their roles.**
(DOCX)

**S1 File. Monte Carlo simulation details.**
(DOCX)

**S1 Fig. PSD-MABP comparison between subjects with and without vasoactive medication.**
Median lowest PSD-MABP levels for subjects with and without vasoactive medication, in all
frequency bands (VLF, LF and HF). No significant differences were found between the two
groups.
(DOCX)

**S2 Fig. ICC values for gain LF and gain VLF.** Beeswarm letter-boxplot with ICC values for
Gain LF (upper figure) and Gain VLF (lower figure) for different cut-off levels of PSD-MABP.
Each analysis method is represented by a letter. No significant differences between methods
were found.
(DOCX)

**S3 Fig. ICC values for phase LF and phase VLF.** Beeswarm letter-boxplot with ICC values
for Phase LF (upper figure) and Phase VLF (lower figure) for different cut-off levels of PSD-
MABP. Each analysis method is represented by a letter. ! :indicates cut-off level at which signif-
icant differences between the methods were found. Post hoc sum-scores for each cut-off level
with significant differences between the methods are indicated in the legend, from left to right.
For each method, a significant positive ICC difference with another method is scored as +1, no
difference as 0, and a negative difference as -1. The sum of all the scores in the post-hoc sum-
score. Negative ICC values do not appear in this plot.
(DOCX)

**S4 Fig. ICC values for ARI and correlation like indices.** Beeswarm letter-boxplot with ICC
values for ARI and correlation like indices for different cut-off levels of PSD-MABP. Each
analysis method is represented by a letter. ! :indicates cut-off level at which significant differ-
ences between the methods were found. Post hoc sum-scores for each cut-off level with signifi-
cant differences between the methods are indicated in the legend, from left to right. For each
method, a significant positive ICC difference with another method is scored as +1, no differ-
ence as 0, and a negative difference as -1. The sum of all the scores in the post-hoc sum-score.
Negative ICC values do not appear in this plot. H and I indicate correlation like indices, which
were not included in calculation of the boxplot or in the statistical analysis.
(DOCX)

**S1 Dataset. All_measurment_files_anonymized.zip: Contains all the anonymized measure-
ment files named 1 through 150.txt.** Allcenters_DCAoutput_andotherdata.xlsx: contains the
DCA analysis output data off all participating centers, Power Spectral Density data for Mean
Arterial Blood Pressure, and means and standard deviations for all variables in the measure-
ment files.
CARNETanonymization.xlsx: contains the anonymization key used for this study.
(ZIP)

## Acknowledgments

The authors wish to thank all participating centers for their efforts and support.

## Author Contributions

**Conceptualization:** Jan Willem Elting, Marit L. Sanders, Ronney B. Panerai, Rong Zhang, Jurgen A. H. R. Claassen.

**Data curation:** Jan Willem Elting.

**Formal analysis:** Jan Willem Elting, Marit L. Sanders, Ronney B. Panerai, Marcel Aries, Edson Bor-Seng-Shu, Alexander Caicedo, Max Chacon, Erik D. Gommer, Sabine Van Huffel, José L. Jara, Kyriaki Kostoglou, Adam Mahdi, Vasilis Z. Marmarelis, Georgios D. Mitsis, Martin Müller, Dragana Nikolic, Ricardo C. Nogueira, Stephen J. Payne, Corina Puppo, Dae C. Shin, David M. Simpson, Takashi Tarumi, Bernardo Yelicich, Rong Zhang, Jurgen A. H. R. Claassen.

**Investigation:** Jan Willem Elting, Ronney B. Panerai.

**Methodology:** Jan Willem Elting, Marit L. Sanders, Ronney B. Panerai, Jurgen A. H. R. Claassen.

**Project administration:** Jan Willem Elting.

**Software:** Jan Willem Elting.

**Supervision:** Jan Willem Elting, Ronney B. Panerai, Jurgen A. H. R. Claassen.

**Validation:** Jan Willem Elting, Marit L. Sanders, Jurgen A. H. R. Claassen.

**Visualization:** Marit L. Sanders.

**Writing – original draft:** Jan Willem Elting, Marit L. Sanders, Ronney B. Panerai, Marcel Aries, Jurgen A. H. R. Claassen.

**Writing – review & editing:** Jan Willem Elting, Marit L. Sanders, Ronney B. Panerai, Marcel Aries, Edson Bor-Seng-Shu, Alexander Caicedo, Max Chacon, Erik D. Gommer, Sabine Van Huffel, José L. Jara, Kyriaki Kostoglou, Adam Mahdi, Vasilis Z. Marmarelis, Georgios D. Mitsis, Martin Müller, Dragana Nikolic, Ricardo C. Nogueira, Stephen J. Payne, Corina Puppo, Dae C. Shin, David M. Simpson, Takashi Tarumi, Bernardo Yelicich, Rong Zhang, Jurgen A. H. R. Claassen.

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
