## [Decision Letter · Decision Letter 0]

7 Nov 2019

PONE-D-19-25218

Assessment of dynamic cerebral autoregulation in humans: is reproducibility dependent on blood pressure variability?

PLOS ONE

Dear Dr. Elting,

Thank you for submitting your manuscript to PLOS ONE. After careful consideration, we feel that it has merit but does not fully meet PLOS ONE’s publication criteria as it currently stands. Therefore, we invite you to submit a revised version of the manuscript that addresses the points raised during the review process.

Three experts raised independent concerns. In the revision all the concerns should be clarified.  Specially clinical implication should be refined.

We would appreciate receiving your revised manuscript by Dec 22 2019 11:59PM. To enhance the reproducibility of your results, we recommend that if applicable you deposit your laboratory protocols in protocols.io, where a protocol can be assigned its own identifier (DOI) such that it can be cited independently in the future. For instructions see: http://journals.plos.org/plosone/s/submission-guidelines#loc-laboratory-protocols

We look forward to receiving your revised manuscript.

Kind regards,

Tatsuo Shimosawa, M.D., Ph.D.

Academic Editor

PLOS ONE

Journal Requirements:

Reviewers' comments:

Reviewer's Responses to Questions

**Comments to the Author**

1. Is the manuscript technically sound, and do the data support the conclusions?

Reviewer #1: Yes

Reviewer #2: Yes

Reviewer #3: Yes

2. Has the statistical analysis been performed appropriately and rigorously? 

Reviewer #1: Yes

Reviewer #2: Yes

Reviewer #3: Yes

3. Have the authors made all data underlying the findings in their manuscript fully available?

Reviewer #1: Yes

Reviewer #2: Yes

Reviewer #3: Yes

4. Is the manuscript presented in an intelligible fashion and written in standard English?

Reviewer #1: Yes

Reviewer #2: Yes

Reviewer #3: Yes

5. Review Comments to the Author

Reviewer #1: This manuscript provides us the novel clinical implications in the assessment of cerebral circulation. The methods were reasonable, and discussion is quite scientific.

To strength the manuscript I would like to ask several minor questions to the authors.

1) How did you calculate the sample size (75 healthy subjects)?

2) As described in the limitation, I also afraid the effects of hypertensive agents to your results. Your discussion about this issue in the limitation paragraph was quite reasonable. If possible, it will be helpful to us that the subjects with hypertensive agents would be excluded or showed in separate table.

3) I would like you to show more concrete clinical perspectives.

Reviewer #2: The reproducibility of dynamic cerebral autoregulation assessment is an important step for its clinical use. The authors found that the assessment was not reproducible unless MAP variations during the measurement window are significant. I found the analysis in the paper is sound. I have the following suggested revisions:

1. The authors mention that reproducibility will probably improve if measurement duration is increased. This is based on their use of the Spearman-Brown prophecy formula. The authors cite a reference for this, but they should also write what the formula is in the manuscript to make it easier to understand Fig. 5. Are longer durations more likely to be reproducible because there are more likely to be significant spontaneous MAP variations?

2. I would appreciate practical recommendations in the manuscript about what the power spectral density MABP levels need to be for reproducible results.

Reviewer #3: The article "Assesment of dynamic cerebral autoregulation in humans: is reproducibility dependent on blood pressure variability?" by Elting et. al. is an interesting and potentially significant article on the intra-sibject variability of cerebral autoregulation measurements. The authors have performed a retrospective analysis of data from several sites, comprising of recordings of mean arterial blood pressure and cerebral blood flow velocity (derived from TCD). The goal of the analysis is to quantify the variability in autoregulation indices specifically with respect to blood pressure variability. The analysis methods, statistical techniques used are rigorous, with instructive and explanative figures. The conclusions drawn by the authors are appropriate. However, a few minor details reduce the clarity and readabilty of the article. Those and other comments are listed below.

1. The organization of the data sources is somewhat difficult to understand. The authors provide a list of data sources and analysis performed therein in supplementary tables S1 and S2. While the text states that data was derived form 5 participating centers, the tables seem to suggest 6 data sources. Furthermore, it is evident that the tables that each data source is used more than once. However, it is difficult for the reader to ascertain if the same class of analysis (e.g., transfer function) is performed more than once on the same data set. Furthermore, are all three categories of analysis avaibale for all data sources? Some clarity on this, perhaps in the form of a supplementary table that lists the data source and associated analysis, will help.

2. One of the goals of the analysis is to look at the variabilities of autoregulatory responses at different time points. However, the authors do not specifiy the range of time pioints from which dual ratings were derived for analysis (besides from an upper bound of 3 months). The time between measurements may be a significant factor in the variablity - this is not considered here.

3. For the surrogate data used to replace data of poor quality, the authors performed simulatons absed on the Tiecks model. How did the authors know which ARI to use, if the data was of poor quality to perform analysis in the first place? Also, please include a citaiton for the model.

4. It might help to describe the outecome variables from the three autoregulation measurement classes.

5. The authors describe the PSD based case removal approach, but do not provide sufficient justification for why the authors considered PSD as a measure of variability. Similarly, what is the rationale for using the 'lowest' PSD-MABP as the independent variable for correlation analyses?

6. PLOS authors have the option to publish the peer review history of their article (what does this mean?). If published, this will include your full peer review and any attached files.

Reviewer #1: Yes: Takuya Kishi

Reviewer #2: No

Reviewer #3: No

---

## [Author Response · Author response to Decision Letter 0]

3 Dec 2019

I attached a separate file with reply's to the reviewers comments

---

## [Decision Letter · Decision Letter 1]

26 Dec 2019

Assessment of dynamic cerebral autoregulation in humans: is reproducibility dependent on blood pressure variability?

PONE-D-19-25218R1

Dear Dr. Elting,

We are pleased to inform you that your manuscript has been judged scientifically suitable for publication and will be formally accepted for publication once it complies with all outstanding technical requirements.

With kind regards,

Tatsuo Shimosawa, M.D., Ph.D.

Academic Editor

PLOS ONE

Additional Editor Comments (optional):

Reviewers' comments:

Reviewer's Responses to Questions

**Comments to the Author**

1. If the authors have adequately addressed your comments raised in a previous round of review and you feel that this manuscript is now acceptable for publication, you may indicate that here to bypass the “Comments to the Author” section, enter your conflict of interest statement in the “Confidential to Editor” section, and submit your "Accept" recommendation.

Reviewer #1: All comments have been addressed

Reviewer #2: All comments have been addressed

Reviewer #3: All comments have been addressed

2. Is the manuscript technically sound, and do the data support the conclusions?

Reviewer #1: Yes

Reviewer #2: Yes

Reviewer #3: Yes

3. Has the statistical analysis been performed appropriately and rigorously? 

Reviewer #1: Yes

Reviewer #2: Yes

Reviewer #3: Yes

4. Have the authors made all data underlying the findings in their manuscript fully available?

Reviewer #1: Yes

Reviewer #2: Yes

Reviewer #3: Yes

5. Is the manuscript presented in an intelligible fashion and written in standard English?

Reviewer #1: Yes

Reviewer #2: Yes

Reviewer #3: Yes

6. Review Comments to the Author

Reviewer #1: (No Response)

Reviewer #2: (No Response)

Reviewer #3: (No Response)

7. PLOS authors have the option to publish the peer review history of their article (what does this mean?). If published, this will include your full peer review and any attached files.

Reviewer #1: Yes: TAKUYA KISHI

Reviewer #2: No

Reviewer #3: No

---

## [Editor Report · Acceptance letter]

30 Dec 2019

PONE-D-19-25218R1 

Assessment of dynamic cerebral autoregulation in humans:  is reproducibility dependent on blood pressure variability? 

Dear Dr. Elting:

I am pleased to inform you that your manuscript has been deemed suitable for publication in PLOS ONE. Congratulations! Your manuscript is now with our production department. 

With kind regards,

on behalf of

Prof. Tatsuo Shimosawa 

Academic Editor

PLOS ONE